# Using *Drosophila melanogaster* to Dissect the Roles of the mTOR Signaling Pathway in Cell Growth

**DOI:** 10.3390/cells12222622

**Published:** 2023-11-14

**Authors:** Anna Frappaolo, Maria Grazia Giansanti

**Affiliations:** Istituto di Biologia e Patologia Molecolari del CNR, c/o Dipartimento di Biologia e Biotecnologie, Sapienza Università di Roma, 00185 Roma, Italy

**Keywords:** Drosophila, mTOR signaling, GOLPH3, Golgi

## Abstract

The evolutionarily conserved target of rapamycin (TOR) serine/threonine kinase controls eukaryotic cell growth, metabolism and survival by integrating signals from the nutritional status and growth factors. TOR is the catalytic subunit of two distinct functional multiprotein complexes termed mTORC1 (mechanistic target of rapamycin complex 1) and mTORC2, which phosphorylate a different set of substrates and display different physiological functions. Dysregulation of TOR signaling has been involved in the development and progression of several disease states including cancer and diabetes. Here, we highlight how genetic and biochemical studies in the model system *Drosophila melanogaster* have been crucial to identify the mTORC1 and mTORC2 signaling components and to dissect their function in cellular growth, in strict coordination with insulin signaling. In addition, we review new findings that involve Drosophila Golgi phosphoprotein 3 in regulating organ growth via Rheb-mediated activation of mTORC1 in line with an emerging role for the Golgi as a major hub for mTORC1 signaling.

## 1. Introduction

Target of rapamycin (TOR) is an evolutionarily conserved serine/threonine kinase, which functions as a central regulator of cellular growth, metabolism and survival in response to environmental cues including nutrients and growth factors [1,2]. Abnormal TOR signaling has been associated with several human diseases including cancer and diabetes [2,3]. The TOR kinase, belonging to the phosphoinositide 3-kinase (PI3K)-related kinase family, owes its name to rapamycin, a macrolide with potent antifungal activity discovered in a soil sample from the island of Rapa Nui (Easter Island) [4]. Subsequent studies of rapamycin revealed its immunosuppressive, antiproliferative and neuroprotective effects [5,6,7]. TOR was originally identified in the budding yeast *Saccharomyces cerevisiae* where mutations in two genes encoding TOR1 and TOR2 proteins confer resistance to the growth-inhibitory properties of rapamycin [8]. It was shown that rapamycin binds the intracellular cofactor propyl-isomerase, FK506 binding protein-12 (FKBP12), to inhibit the activities of TOR1 and TOR2 proteins [8]. Unlike yeast, most other species including humans and *Drosophila melanogaster* harbor only one gene encoding TOR, which is the catalytic subunit of two distinct functional multiprotein complexes termed mTORC1 (mechanistic target of rapamycin complex 1) and mTORC2 ([3,9]; Figure 1).

Both complexes share the TOR kinase subunit and the mammalian homologue of LST8 (mLST8, lethal with SEC13 protein 8) [1,10]. However, they exhibit differential sensitivity to rapamycin and contain accessory proteins that are unique to each complex (Figure 1). The regulatory-associated protein of mTOR (RAPTOR; [10,11]), the proline-rich Akt substrate 40 kDa (PRAS40; [12,13]) and DEPTOR (DEP domain containing mTOR interacting protein; [14]) are associated with mTORC1. Conversely, mTORC2 is defined by the scaffolding protein rapamycin-insensitive companion of mTOR (RICTOR), which recruits the mammalian stress-activated MAP kinase interacting protein 1 (mSIN1; [15,16,17]), protein-associated with rictor 1 or 2 (PROTOR1/2; [18,19]) and DEPTOR [14] to form the complex. Many studies indicate that mTORC1 and mTORC2 play different physiological functions and have distinct substrates. mTORC1 controls cell growth and promotes protein translation by phosphorylating the downstream targets ribosomal p70 S6 kinase I (S6K) and the ribosome-associated eukaryotic translation initiation factor 4E-binding protein (4E-BP; [20,21,22]). On the other hand, mTORC2 regulates cytoskeletal remodeling, cell proliferation and survival by regulating the activity of members of the AGC (PKA/PKG/PKC) family of protein kinases [18,19,23,24,25,26]. Unlike mTORC1, mTORC2 was originally shown to maintain the ability to phosphorylate its substrates upon acute treatment with rapamycin and described as a rapamycin-insensitive mTOR complex ([18,19], Figure 1). However, it has been reported that prolonged treatment with rapamycin can impair mTORC2 integrity and activity by sequestering mTOR into rapamycin-bound complexes [27,28]. 

In recent years, many research studies and review articles have been focused on the new roles of TOR signaling in aging processes in *Drosophila melanogaster* and other model organisms [29,30,31,32,33,34,35,36,37]. Here, we review how research studies on *Drosophila melanogaster* have significantly contributed to our understanding of the mTOR signaling pathway and its involvement in cellular growth.

## 2. mTORC1 Signaling in *Drosophila melanogaster* Regulates Organ Growth

Extensive studies in genetically tractable organisms including *Drosophila melanogaster* have greatly contributed to understanding the intricate mTORC1 molecular pathway [9]. As a central cell growth regulator, mTORC1 responds to a variety of upstream signals including growth factors and the availability of amino acids and glucose [9]. mTORC1 is a downstream effector of growth factors such as insulin or insulin-like growth factors through the insulin receptor (InR)/phosphoinositide 3-kinase (PI3K)/protein kinase B (Akt) signaling pathway [38]. Following the binding of insulin or insulin-like growth factors (IGFs) to their receptors and phosphorylation of the insulin receptor substrate (IRS), PI3K is engaged by IRS in the cell membrane and converts phosphatidylinositol-4,5-biphosphate (PIP2) to phosphatidylinositol-3,4,5-triphosphate (PIP3). The PTEN tumor suppressor is a negative regulator of this pathway, acting as a phosphatase to convert PIP3 to PIP2 (for review, see [39]). In turn, PIP3 accumulation recruits and activates the serine-threonine kinase Akt (also known as protein kinase B) to the plasma membrane via phosphorylation mediated by the phosphoinositide-dependent protein kinase 1 (PDK1) and mTORC2 [9].

The PI3K/Akt pathway converges to mTORC1 via the key signaling regulator known as the tuberous sclerosis complex (TSC, [40,41,42]; Figure 1). The TSC complex, composed of TSC1 and TSC2 proteins, negatively regulates the mTORC1 pathway by functioning as a GTPase-activating protein (GAP) and a negative regulator of the GTPase protein Rheb (Ras homolog enriched in brain). The GTP-bound form of Rheb directly interacts with the kinase domain of mTOR and strongly stimulates its kinase activity [43,44,45,46,47]. AKT-mediated phosphorylation of TSC2 (also known as tuberin) disrupts the TSC complex, relieving its inhibitory effects on mTORC1 signaling [38,41,42,48]. The Tre2-Bub2-Cdc16 (TBC) 1 domain family, member 7 (TBC1D7) was recently identified as the third and stoichiometric component of the TSC complex, which stabilizes the association of TSC1 and TSC2 proteins by directly interacting with the TSC1 protein [40]. The near-atomic resolution of the human TSC complex by cryoelectron microscopy (cryo-EM) indicates that it has an arch-shaped structure with a 2:2:1 stoichiometry of TSC2 to TSC1 to TBC1D7 [49].

Genetic and biochemical characterization of the Drosophila homologs of the mTORC1 signaling components has illuminated the role of the mTOR pathway in organ growth and its relationship with the insulin pathway. Studies from two different research groups analyzed the phenotypes associated with mutations in the unique Drosophila *TOR* (*dTOR*) encoding gene, revealing striking similarities with the effects of nutrient deprivation or mutations in insulin signaling [50,51]. Mutational inactivation of the insulin/PI3K/Akt signaling pathway in Drosophila is associated with a decrease in cell size and organ growth accompanied by an extended G1 phase of the cell cycle, developmental delay and defective cell proliferation [41,50,51,52,53,54,55,56,57,58,59,60,61,62]. Like PI3K pathway mutants, animals carrying *dTOR* mutant alleles display defects in cell cycle progression and cell growth in several tissues during Drosophila development [51]. Loss of dTOR in Drosophila larvae also causes several cellular phenotypic defects that can be observed in amino-acid-starved wild-type larvae, including reduced nucleolar size, lipid vesicle aggregation in the larval fat body and a cell-type-specific growth arrest [51]. In imaginal disc cells from *dTOR* mutant larvae, the length of the G1 phase of the cell cycle is extended while the level of the G1/S-phase regulator cyclin E (CycE) is reduced. Moreover, the cell cycle arrest of *dTOR* mutant cells can be rescued by overexpression of CycE indicating that TOR signaling regulates cell proliferation by maintaining normal rates of CycE [51]. As in mammalian cells, dTOR is required for growth factor-dependent phosphorylation of activated Drosophila S6K (dS6K) and overexpression of dS6K *in vivo* can partially rescue the phenotypes of *dTOR* loss of function mutants [51]. The Drosophila genome harbors a single *dS6K* gene whereas the two orthologs *RPS6KB1* and *RPS6KB2* genes in humans encode the S6K1 and S6K2 proteins, respectively [63,64,65,66,67]. Most flies carrying null alleles of Drosophila *S6K* (*dS6K*) die before reaching the adult stage; the surviving flies are smaller than wild-type flies [68]. Notably, eye and wing tissues from mutant flies lacking *dS6K* exhibit a reduced cell size but a cell number that is comparable to wild-type tissues [68].

Characterization of mutations in the Drosophila *Tsc1* (*dTsc1*) and *dTsc2*/*gigas* genes revealed that the heterozygosity can suppress the lethality associated with loss-of-function insulin receptor mutants [59]. Imaginal disc cells carrying mutations in either dTsc1 or dTsc2 show enhanced growth and increased size but retain normal ploidy [69]. Later studies demonstrated that Drosophila Tsc1 and Tsc2 proteins form a complex and antagonize insulin signaling to regulate cellular and organ size and function upstream of TOR [70]. Loss of *dTsc1* and *dTsc2* leads to a TOR-dependent increase in dS6K activity and inhibition of Drosophila Akt, with the latter effect that can be suppressed by loss of dS6K [70,71]. By using both *in vitro* and *in vivo* assays, Zhang and coauthors [47] provided evidence that the Drosophila small GTP-ase Rheb (dRheb) is the direct target of the Tsc2 GAP activity. Moreover, point mutations in the GAP domain of the Tsc2 protein disrupt the GAP activity towards dRheb. Consistent with these results, other studies in the same year showed that dRheb regulates cell growth, downstream of dTsc1-dTsc2 and upstream of dTOR in the insulin-dTOR signaling pathway [45,46,72]. In humans, somatic loss of heterozygosity in either the *TSC1* or *TSC2* tumor suppressor gene causes tuberous sclerosis, a disease characterized by the frequent development of benign tumors in various organs [73,74]. Although TBC1D7 has been reported as a core component of the TSC complex, loss-of-function mutations in *TBC1D7* have not been reported in TSC patients [40]. However, germline homozygous truncating mutations in *TBC1D7* have been associated with a rare autosomal-recessive megalencephaly and neurocognition syndrome [75,76]. These divergent phenotypes prompted Ren and coauthors [77] to use *Drosophila melanogaster* as a model system to investigate the physiological roles of Drosophila TBC1D7 (dTBC1D7) *in vivo*. In Drosophila, the major insulin-like peptides, ILP2, ILP3, and ILP5, are expressed in a set of median neurosecretory cells of the fly brain, known as insulin-producing cells (IPCs) and released in the hemolymph [78,79]. By using CRISPR/Cas9-mediated genome editing, Ren and coauthors [77] generated a null *dTBC1D7* mutant. In contrast with *TSC1* and *TSC2* loss of function mutants, which are lethal, mutants lacking *dTBC1D7* are viable, suggesting that the dTBC1D7 protein is not a constitutive component of the TSC. Indeed, they showed that dTBC1D7 is expressed in the IPCs of the fly brain and controls systemic growth in a cell-nonautonomous and TSC-independent manner, by selectively regulating the expression and release of ILP2. 

Translationally controlled tumor protein (Tctp) is a highly conserved protein which plays an essential role in regulating cell growth and has been involved in tumorigenesis and tumor reversion [80,81,82]. Genetic and biochemical analyses in Drosophila have identified Tctp as a molecular component of the TSC-Rheb pathway required for controlling both the cell size and cell number in imaginal discs [83,84]. Work from Hsu and coauthors [83] showed that the Drosophila TCTP (dTctp) protein directly binds dRheb and functions as a guanine nucleotide exchange factor for it in both *in vivo* and *in vitro* assays. The cell growth defects associated with down-regulation of dTctp can be rescued by human TCTP (hTCTP), suggesting that the function of TCTP in the Rheb-mTORC1 signaling is evolutionarily conserved [83]. Consistent with this hypothesis, another study provided evidence *in vitro* that hTCTP can accelerate the GDP release of the human Rheb protein in biochemical assays [85]. 

The functional role of Drosophila PRAS40 (dPRAS40) in coupling insulin and mTORC1 signaling was investigated *in vivo* in flies carrying a loss-of-function mutant allele [86]. Biochemical and genetic analyses demonstrate that the dPRAS40 function regulates TORC1 activity and links insulin signaling to TORC1 in Drosophila [86]. However, unexpectedly, although PRAS40 is expressed in all tissues, it regulates TORC1 activity in ovaries, but not in other tissues, thereby influencing fertility but not animal growth. In mammals, 14-3-3 proteins were shown to regulate TOR signaling by binding several components of the mTORC1 complex including PRAS40 [87]. The Drosophila genome harbors two genes encoding 14-3-3 proteins, named *14-3-3 ε* and *14-3-3 ζ*, respectively [88]. In Drosophila 14-3-3 ε and 14-3-3 ζ proteins physically interact with both dTctp and dRheb proteins [84]. Single knockdown of either Drosophila *14-3-3 ε* and *14-3-3 ζ* in the wing or eye imaginal discs does not impair the wing/eye growth but causes genetic interaction with *dTctp* and *dRheb* to regulate organ growth. Moreover, double knockdown of *14-3-3 ε* and *14-3-3 ζ* disrupts organ growth and mTORC1 signaling by affecting the interaction between Tctp and Rheb [84].

A recent integrative analysis combined affinity purification-mass spectrometry with RNA interference screening and phosphoproteomic data, to identify new molecular targets of the InR/PI3K/Akt pathway in Drosophila [89]. This study revealed that approximately 10% of the interacting proteins are dynamically phosphorylated upon insulin stimulation. The Chaperonin containing TCP-1 (CCT) complex subunit CCT8 is one of the new molecular targets of insulin-dependent phosphorylation, identified during this proteomic analysis. The CCT complex has been predicted to control protein folding of about 10% of newly synthesized cytoplasmic proteins [90]. Consistent with the involvement of the CCT complex in InR/PI3K/Akt signaling, Kim and Choi [91] demonstrated that the Drosophila CCT complex physically interacts with dTOR, dRheb and dS6K proteins. Moreover, loss of the CCT complex results in severe cell growth defects during organ development and affects the level of phosphorylated dS6K while the total level of dS6K remains unchanged. Finally, the TOR signaling pathway regulates the transcription of the CCT complex. Overall, these results suggest that TOR-mediated control of cell growth also depends on the protein folding machinery.

## 3. Dissecting the Role of mTORC2 in Cell Growth in *Drosophila melanogaster*

In contrast to mTORC1, the mTORC2 signaling network is still poorly characterized. To date, mTORC2 is known to phosphorylate the following substrates belonging to the AGC kinase family: the oncogene Akt [92], several classes of protein kinases C (PKCs; [18,26,93]) and the ion transport regulator serum- and glucocorticoid-induced protein kinase 1 (SGK1; [23]). The mTORC2 kinase activity is primarily responsive to growth factor-mediated stimulation [3,94]. As previously mentioned, insulin or insulin-like growth factors activate the PI3K pathway, resulting in the accumulation of PIP3 to the plasma membrane. Two alternative models illustrate the upstream regulation of mTORC2 [94]. Liu and coauthors [95] propose that mTORC2 is directly activated by the level of PIP3 in the plasma membrane. According to this model, binding of the mSIN1 pleckstrin homology (PH) domain to PIP3 leads to mTORC2 recruitment to the plasma membrane releasing mTOR inhibition. On the other hand, a recent work by Ebner et al. [96] suggested that a pool of mTORC2 is localized to the plasma membrane and is constitutively active. Moreover, extensive work in several model systems has shown that mTORC2 activation is mediated by the small GTPases Rab35, Ras, Rac1 and Rap1 in a PI3K-dependent or -independent manner [97,98,99,100,101]. In addition to growth factors and PI3K-mediated activation, mTORC2 also responds to amino acid levels [102,103] and to other signaling pathways including AMP-activated protein kinase (AMPK; [104,105,106]), and Wingless/Integrated (Wnt) signaling [107,108]. On the other hand, mTORC2 is negatively regulated by mTORC1 in a feedback control loop [109]. Indeed, mTORC1 inhibits mTORC2 through downregulation of insulin signaling by both direct phosphorylation of the negative regulator Grb10 (growth factor bound-receptor protein 10; [110,111]) and S6K phosphorylation of IRS1 [112,113].

The Drosophila genome encodes proteins homologous of all the mTORC2 complex components. Moreover, *Drosophila melanogaster* has been extensively exploited as a model system for cellular and *in vivo* studies aimed at dissecting the mTORC2 pathway. RICTOR was independently identified in human and Drosophila S2 cells by two distinct groups [18,19]. Knockdown of both human and Drosophila RICTOR (dRictor) proteins was demonstrated to reduce the phosphorylation on Ser473 (Ser505 in Drosophila) of the hydrophobic motif of Akt, while the levels of phosphorylated S6K remained unchanged [18,92]. These data led to the identification of Rictor-mTOR as the PDK2 complex required for phosphorylation of Akt on Ser473. PDK2/mTORC2 also facilitates phosphorylation on Thr308, which is mediated by PDK1 [114,115], allowing the complete PI3K-dependent activation of Akt at the cell membrane [92]. In addition, RNA interference (RNAi)-mediated knockdown in S2 cells demonstrated that Drosophila Sin1 (dSin1) is a key component of the mTORC2 complex required for Akt activation [17]. The great advantages offered by the sophisticated genetic tools in Drosophila have been essential in analyzing the *in vivo* role of mTORC2. Indeed, it is worth noting that, in mice, depletion of each mTORC2 component leads to embryonic lethality [16,17,116,117]. Besides the *dTOR* mutant (described above), mutants in *dRictor* and *dSin1* have also been generated, both of which are homozygous viable [118,119]. *dRictor* and *dSin1* mutant flies are smaller than controls and exhibit a significant general reduction in body size and tissue growth, as demonstrated by the small-wing phenotype [120,121]. Furthermore, genetic interaction experiments demonstrated that a null mutation in *dRictor* counteracts the increase of the eye area and of the ommatidia size caused by Akt overexpression [121]. Akt is known to phosphorylate and inhibit the TSC complex allowing mTORC1 activation (see above, [38,41]) but it can also promote cell survival by inhibition of the forkhead-box FOXO1/3a transcription factors [3]. Akt and SGK-mediated phosphorylation of FOXO1/3a is insulin-dependent and leads to FOXO1/3a retention into the cytoplasm via binding to 14-3-3 proteins [120,121]. Overexpression in the developing eye of Drosophila Foxo (dFoxo), the only Drosophila FOXO1/3a ortholog, results in increased levels of apoptosis and a significant reduction and roughness of the eye accompanied by loss of ommatidia [118,119,122,123,124]. The eye phenotype, associated with dFoxo overexpression, is strongly enhanced in a *dRictor* mutant background [118,119]. Overall, these data have led to the proposal that mTORC2 positively regulates *in vivo* cell growth and survival through Akt/FOXO signaling. 

mLST8/Lst8, together with TOR, is a common subunit of both mTORC1 and mTORC2 complexes. Unlike Drosophila mutants in the *dTor* or *dRheb* genes, which encode essential components of mTORC1 and are homozygous lethal, flies carrying null alleles in Drosophila *Lst8* (*dLst8*) are viable [125]. However, *dLst8* mutant flies exhibit a general reduction in tissue growth, a phenotype also observed in *dRictor* loss-of-function mutants and characterized by a smaller body size and smaller wings and eyes. By using two experimental genetical approaches in Drosophila, Wang and coworkers demonstrated a cell-autonomous role for *dLst8* in the growth in different cell types [125]. In genetic mosaics, cells lacking dLst8 were smaller than neighboring wild-type cells. In addition, expression of dLst8 protein in the posterior compartments of developing wings of *dLst8* mutants specifically rescued the growth defects of the posterior cells but not the anterior. Work in both Drosophila and mammalian cells indicates that mLST8/dLst8 is dispensable for the mTORC1 function but essential for the mTORC2 activity [117,125]. In a *dLst8* mutant background, the overexpression of dRheb can drive upregulation of mTORC1, promoting cell growth. Moreover, in *Drosophila melanogaster* loss of dLst8 does not affect the phosphorylation of dS6K, a major target of mTORC1, whereas it largely reduces the phosphorylation of Akt on Ser505, which is the direct target site of mTORC2. Although mTORC2-dependent-Akt phosphorylation is compromised in *dLst8* mutant flies, the expression of a constitutively active, phospho-mimetic form of Akt cannot rescue the cell growth defects associated with loss of dLst8 [125]. It has been suggested that phosphorylation of Akt by mTORC2 inactivates FOXO signaling without affecting the TSC2-mTORC1 branch [16,17,117]. In agreement with this hypothesis, loss of either dRictor or dLst8 exacerbates the phenotype associated with FOXO overexpression [118,125]. However, mutations in *dFoxo* do not increase organ growth in the wild type and do not compensate for the defects in organ growth caused by lack of dLst8. The unexpected results in this study suggest that Akt signaling alone cannot mediate mTORC2 effects on cell growth, indicating the need to identify new substrates of mTORC2. In this context, a more recent work by Kuo et al. [126] characterized the role of the transcription factor Myc in cell growth in Drosophila, showing that mTORC2 and Myc act in a common pathway. Epistasis analysis showed that the cellular and tissue growth defects in animals carrying a loss-of-function Drosophila *Myc* (*dMyc*) allele are not exacerbated by loss of *dLst8*. In contrast to the effects of Akt overexpression [125], overexpression of Drosophila Myc fully suppresses the growth defects caused by either *dLst8* or *dRictor* mutations [126]. In addition, dLst8 and dRictor control the nuclear localization of dMyc and, thus, Myc-dependent transcription of a large set of growth-related genes. Kuo and coworkers propose that the mTORC2/Myc pathway regulates the expression of several classes of proteins required for metabolic and catabolic processes, that are all strictly linked to cell growth [126]. 

## 4. Golgi Phophoprotein 3 (GOLPH3) Controls Organ Growth in Drosophila by Directly Associating with mTOR Signaling Proteins

The most accredited model posits that mTORC1 is recruited to the lysosomal surface, where it is activated by GTP-bound Rheb [127,128]. Consistent with this model, the lysosome functions as an important hub for activation of mTOR in response to amino acids, a process that requires the Rag family GTPases and the so-called Ragulator [129,130]. In mammalian cells, the RagA/RagB or RagC/RagD stable heterodimers are recruited to the lysosomal membrane by the Ragulator complex [130]. Upon amino acid stimulation, Rags are converted to their nucleotide-active state to mediate cellular translocation of the mTORC1 complex to the lysosomes [130]. However, many recent reports have localized the mTOR kinase and its activator Rheb to the Golgi apparatus and provided evidence that the Golgi apparatus could be directly involved in mTORC1 activation [131,132,133,134]. In a recent study, the active-phosphorylated form of the mTOR kinase was also located on Golgi membranes of HeLa cells, and a proper Golgi ribbon architecture was shown to be essential for functional regulation of mTOR activity [132]. Another research article reported Rheb localization in the Golgi of HeLa cells as well as a novel inter-organelle contact site, between the Golgi and lysosome involved in mTORC1 activation [135].

In agreement with a role for the Golgi apparatus in regulating TOR signaling, the oncogenic activity of human Golgi phosphoprotein 3 (GOLPH3) correlates with increased growth factor-induced mTOR signaling [136,137]. Overexpression of human GOLPH3 has been associated with poor prognosis in multiple solid tumors including breast cancer, colon cancer and glioblastoma [136,137,138]. However, the molecular mechanisms whereby GOLPH3 modulates the mTOR-pathway remain elusive [137]. GOLPH3 is a highly conserved phosphatidylinositol 4-phosphate (PI4P) binding protein, that is enriched in the trans-Golgi network and involved in vesicle trafficking and Golgi structure maintenance in human cells and *Drosophila melanogaster* [139,140,141,142]. 

We have recently demonstrated that GOLPH3 controls organ growth in *Drosophila melanogaster* by directly interacting with the Akt/TOR signaling proteins ([143], Figure 2).

dGOLPH3 physically associates with Tctp and 14-3-3 ζ and dLst8 proteins. Wild-type function of Drosophila GOLPH3 (dGOLPH3) is required to control tissue and organ growth during fly development in the Tctp-Rheb-mTORC1 axis. Knockdown of *dGOLPH3* in the wing or eye imaginal discs reduces the fly wing/eye size, mimicking the *dTctp*-mutant phenotypes. Genetic and epistasis analyses showed that dGOLPH3 acts upstream of Tctp and 14-3-3 ζ in the mTOR growth control machinery. Importantly, we showed that the dRheb protein depends on dGOLPH3 for its recruitment to the Golgi apparatus. Two non-conservative mutations in the PI4P binding pocket of the dGOLPH3 protein that impair dGOLPH3 localization to Golgi membranes also disrupt its interaction with the dRheb protein, suggesting that the two molecular partners may associate at the Golgi apparatus. These findings are in line with a role for the Golgi complex as a hub for mTOR activation [131,132,133,134]. dGOLPH3 also promotes formation of the dRheb-dTctp complex that supports activation of mTORC1. In this context, dGOLPH3 directly binds 14-3-3 ζ and dTctp. However, it remains to be investigated whether dGOLPH3 engages the other mTOR signaling components at the Golgi apparatus or in other cellular locations. Consistent with dGOLPH3 involvement in Rheb-mediated mTORC1 activation, depletion of dGOLPH3 also affects the levels of phosphorylated ribosomal dS6 kinase, a downstream target of mTORC1. 

Our finding that dGOLPH3 interacts with dLst8 suggests that dGOLPH3 may also control mTORC2. In agreement with this hypothesis, upregulation of human GOLPH3 been associated with enhanced activation of both mTORC1 and mTORC2 complexes [136]. Importantly, although mLST8/dLst8 is the only conserved TOR-binding protein that is shared by both mTORC1 and mTORC2, its function is essential for mTORC2 activity but dispensable for mTORC1 function in both Drosophila and mammalian cells (see above [117,125,144]).

## 5. Conclusions and Perspectives

Experiments in animal models and analysis of the clinical database of cancer patients provide evidence that mTOR and its upstream regulators contribute to tumorigenesis [145,146]. Indeed, hyperactivation of mTOR signaling has been reported in more than 70% of cancers [147]. *Drosophila melanogaster* has emerged as a powerful model system to identify the molecular components of the TOR signaling and to study their functional role *in vivo* in cellular growth regulation. Thus, further analysis of this pathway in flies will help identify new molecular targets in the development of therapeutic strategies for precision medicine in cancer. Although the oncogenic activity of human GOLPH3 has been linked to enhanced signaling downstream of mTOR, the GOLPH3-dependent molecular pathways leading to malignant transformation remain to be clarified. Our recent findings have provided the first *in vivo* demonstration that the oncoprotein GOLPH3 controls tissue growth in Drosophila by directly associating with the TOR signaling proteins and controlling Rheb-mediated activation of mTORC1 at the Golgi [143].

Importantly, recent studies have identified the small GTPase Rab1A, a component of the Golgi trafficking machinery, as a key regulator of the mTORC1 pathway and of cell growth [131]. Moreover, the activation of mTORC1 by Rab1A at the Golgi was shown to be independent of mTORC1 activation at the lysosomes, which depends on Rag small GTPases. Rab1A regulates the formation of the Rheb-mTORC1 complex in the Golgi by interacting with the mTORC1 subunit Raptor [131]. Human Rab1A is overexpressed in colorectal carcinoma (CRC), which is correlated with enhanced mTORC1 signaling and poor prognosis [131]. Likewise, overexpression of GOLPH3 is significantly associated with increased mTORC1 signaling and poor clinical outcome in colorectal cancer patients [136,137]. Because GOLPH3 binds and behaves as an effector of Rab1 in both Drosophila and human cells [141,148], the Drosophila genetic toolkits and assays can be used to test whether GOLPH3 and Rab1 function in the same pathway to regulate mTORC1 activation in the Golgi apparatus. It will also be important to investigate the interplay of GOLPH3/dGOLPH3 with mLST8/dLst8 in mTORC2 signaling. A recent study in a panel of normal and cancer cells showed that mLST8 binds with the SIN1-RICTOR complex and acts as a scaffold protein for the assembly and kinase activity of mTORC2. Lack of mLST8 or mutations in the mLST8 sequences required for mTOR binding selectively block the association of mTOR with the mTORC2 cofactors RICTOR and SIN1 without interfering with mTORC1 in multiple untransformed and cancer cell lines. [144]. Moreover, in *PTEN*-null prostate cancer xenografts, mutations in the mTOR interaction motif of mLST8 specifically affect mTORC2 activity and decrease tumor growth *in vivo*. Overall, these data suggest that drug design strategies and the targeting of mLST8–mTOR interactions may be effective for mTORC2-selective inhibition in cancer therapy [144]. Notably, *Drosophila melanogaster* has emerged not only as a suitable model organism for modeling several types of human tumors, but also as a valuable platform for high-throughput screening of anticancer drugs and personalized therapies [149].

## Figures and Tables

**Figure 1 cells-12-02622-f001:**
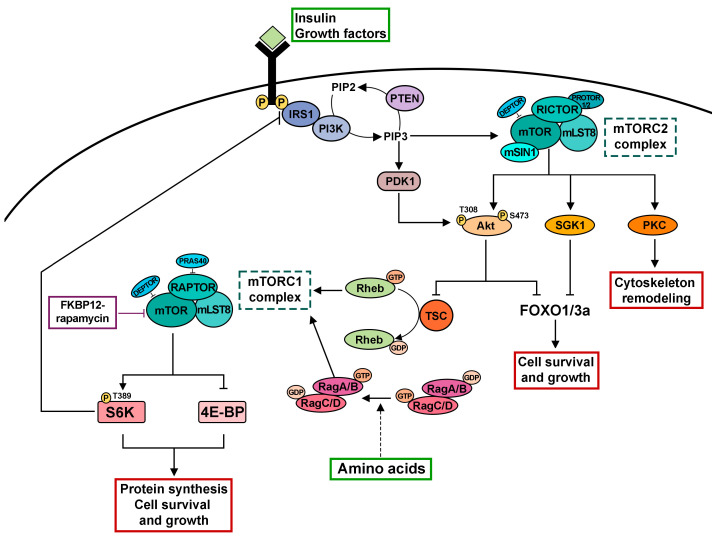
Cartoon depicting the mTOR signaling network. The mTOR kinase is the catalytic subunit of two distinct functional multiprotein complexes termed mTORC1 and mTORC2. mTORC1 and mTORC2 are characterized by a differential sensitivity to rapamycin and distinct accessory proteins. mTORC2 responds to growth factors by phosphorylating and activating several AGC family kinases including Akt, SGK1 and PKC, thereby regulating cell survival, cell growth and cytoskeletal dynamics. After binding of insulin or IGFs to their receptors and phosphorylation of the IRS1, PI3K is recruited to the cell membrane, leading to the accumulation of PIP3. PTEN functions as a negative regulator of this pathway converting PIP3 to PIP2. In turn, PIP3 triggers the recruitment of Akt to the plasma membrane followed by its activation via phosphorylation mediated by PDK1 and mTORC2. Akt phosphorylates the TSC2 subunit of the TSC complex to inhibit its GAP activity for the small GTPase Rheb, the direct activator of mTORC1. Amino acids regulate mTORC1 in a process that depends on the Rag family GTPases. A key function of mTORC1 is to promote protein synthesis for cell growth by phosphorylating S6K and 4E-BP. The following abbreviations are used: mTO RC1/mTORC2, mechanistic target of rapamycin complex 1/mechanistic target of rapamycin complex 2; FKBP12, propyl-isomerase, FK506 binding protein-12; LST8, lethal with SEC13 protein 8; RAPTOR, regulatory-associated protein of mTOR; PRAS40, proline-rich Akt substrate 40 kDa; DEPTOR, DEP domain containing mTOR interacting protein; RICTOR, rapamycin-insensitive companion of mTOR; mSIN1, mammalian stress-activated MAP kinase interacting protein 1; PROTOR1/2, protein-associated with rictor 1 or 2; AGC, cAMP-dependent, cGMP-dependent and protein kinase C; Akt, Protein kinase B; SGK1, serum- and glucocorticoid-induced protein kinase 1; PKC, Protein kinase C; IGFs, insulin-like growth factors; IRS1, insulin receptor substrate; PTEN, Phosphatase and TENsin homolog deleted on chromosome 10; PIP3, phosphatidylinositol 3,4,5-trisphosphate; PIP2, phosphatidylinositol 4,5-Bisphosphate; PDK1, Phosphoinositide-dependent protein kinase-1; TSC, tuberous sclerosis complex; Rheb, Ras Homolog Enriched in Brain 1; S6K, ribosomal p70 S6 kinase I; 4E-BP, eukaryotic translation initiation factor 4E-binding protein.

**Figure 2 cells-12-02622-f002:**
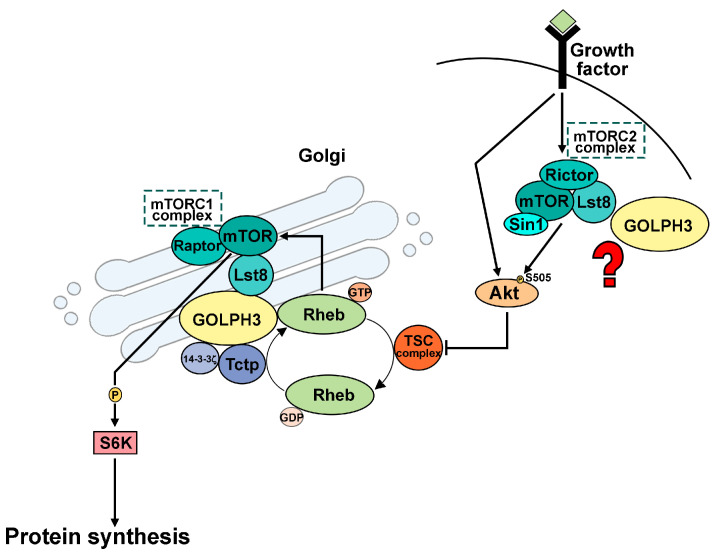
Schematic illustrating the role of Drosophila GOLPH3 in the mTOR signaling. mTORC1-mediated phosphorylation of S6K stimulates protein synthesis for cell growth. Growth factors like insulin activate PI3K, followed by activation of Akt through phosphorylation mediated by PDK1 and mTORC2. The small GTPase Rheb is an essential activator of mTOR and the TSC complex functions as a GAP for Rheb. Akt is known to phosphorylate the TSC2 subunit of the TSC complex to inhibit its GAP activity. Tctp regulates organ growth by forming a complex with 14-3-3 proteins and Rheb and acts as a GEF for Rheb. GOLPH3 protein controls Rheb localization to the Golgi membranes and facilitates Tctp–Rheb interaction. GOLPH3 also interacts with the Lst8 protein, suggesting an additional role for this protein in regulating mTORC2 activity.

## Data Availability

Not applicable.

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
