# Peer review of "Using Drosophila melanogaster to Dissect the Roles of the mTOR Signaling Pathway in Cell Growth"

_cells, 2023, doi:10.3390/cells12222622_

Round 1

Reviewer 1 Report

Comments and Suggestions for Authors

This work by Frappaolo and Giansanti reviews mTOR signaling for cell growth and survival. It focuses on the function of two distinct but related complexes, mTORC1 and mTORC2. In particular, this review outlines the contribution of the Drosophila model research in dissecting the in vivo function of mTOR signaling in organ growth.

Section 1 (Introduction) describes a model for the mTOR signaling network in which insulin and insulin-like growth factors regulate PI3 kinase to activate mTORC2. The model illustrates how the mTORC1 complex is regulated by the Akt-Tsc-Rheb pathway.

In sections 2 and 3, the authors explain more details about mTORC1 and mTORC2 signaling in Drosophila. The roles of various components of these complexes are summarized.

Section 4 provides interesting recent studies on the role of GOLPH3, a Golgi-associated phosphoprotein. It discusses the proposal that GOLPH3 is involved in the localization of the Rheb-Tctp complex to activate TORC1 signaling at Golgi.

Overall, it is a well-written review of Drosophila model studies on mTOR signaling. I have a few minor comments:

1.     The authors did an excellent job discussing how the Drosophila model has contributed to the current knowledge of mTOR signaling. However, only about 15% of cited papers were published in the last 5 years, indicating that the authors emphasized relatively older and historically significant papers. Given the numerous recent studies on TOR signaling in Drosophila and other systems (new roles in aging, interaction with different signaling pathways, role of Golgi, etc.), I suggest the authors include more recent studies. Note that the Journal recommends most citations in the past 5 years.

2.     Line 55-69 (“The following abbreviations are used …”.). This paragraph for the abbreviations is described as part of the Introduction. The abbreviations should be separated from the Introduction. They may be included in the Figure 1 legend or in a separate box.

3.     Line 70-83, line 114. Full names of several TOR components (for example, LST8, DEPTOR, mSIN1, etc) are better removed since they are already listed in the abbreviations.  

4.     ‘Drosophila melanogaster’: Both plain and italic fonts are used. Italicize all of them. 

Author Response

Point-by-point responses to reviewers’ comments.

(The reviewers’ comments are in Italics. Our responses are non-italicized)

Reviewer #1

  1. The authors did an excellent job discussing how the Drosophila model has contributed to the current knowledge of mTOR signaling. However, only about 15% of cited papers were published in the last 5 years, indicating that the authors emphasized relatively older and historically significant papers. Given the numerous recent studies on TOR signaling in Drosophila and other systems (new roles in aging, interaction with different signaling pathways, role of Golgi, etc.), I suggest the authors include more recent studies. Note that the Journal recommends most citations in the past 5 years.

To meet the reviewer’s request, we have now cited recent research articles focused on the new roles of TOR in aging processes in Drosophila melanogaster and other model organisms (Lines 90-92 and Lines 501-521). Moreover, we have included more recent studies on TOR signaling in cell growth in Drosophila (Lines 209-223 and Lines 638-645) and on the role of Golgi (Lines 334-336 and Lines 746-747).

  1. Line 55-69 (“The following abbreviations are used ...”.). This paragraph for the abbreviations is described as part of the Introduction. The abbreviations should be separated from the Introduction. They may be included in the Figure 1 legend or in a separate box.

We have now included the abbreviations in the Figure 1 Legend (Lines 55-67).

  1. Line 70-83, line 114. Full names of several TOR components (for example, LST8, DEPTOR, mSIN1, etc) are better removed since they are already listed in the abbreviations.

Because the abbreviations have been included in the Figure 1 Legend, we did not remove full names of TOR components in the introduction, to facilitate the reading.

  1. ‘Drosophila melanogaster’: Both plain and italic fonts are used. Italicize all of them.

Drosophila melanogaster” is now in italics all over the manuscript.

Reviewer 2 Report

Comments and Suggestions for Authors

This review article is nicely written. This reviewer feels the authors must address only a few questions and modify minor mistakes.

1.  In lines 259-261, these data have suggested that TORC2 positively regulates cell growth and survival through Akt/FOXO signaling. In contrast, in line 284-285, these data indicate that the effects of TORC2 on cell growth are not mediated by its target Akt and FOXO signaling. There is an inconsistency between these two sentences. Please describe the author’s views on these points.

2.  Lines 303-305, from the original expression, I feel that amino acids stimulation facilitates formation of Rag heterodimer, instead of the promotion of changes of GTP-GDP cycles of each Rag GTPases. Please rewrite.

3.  Lines 170-177, the authors should briefly summarize the molecular function of TBC1D7. Is TBC1D7 not required for TSC in Drosophila?

4.  Throughout the manuscript, the authors use mTORC1 and TORC1 (mTORC2 and TORC2). I understand that mTORC1 and mTORC2 may be used only for mammals. However, for example, in line 203, the authors used “mTORC1”, but it should be TORC1 because the original experimental data were obtained in Drosophila. Most readers might be confused. Please add the definition of how the terms are used, for example, mTORC1 for mammals, and TORC1 for other species such as Drosophila. Please confirm the use of these terms throughout the manuscript.

5.   Lines 55-69, Please modify.

6.  Line 82, S6K1 is used, but S6K in Figure 1. Please modify the expression consistently.

7.  Line104, full name of PIP2 and PIP3 should be --- bisphosphate and --- trisphosphate, respectively.

8.  Line 108, PDPK1 should be PDK1.

Author Response

Point-by-point responses to reviewers’ comments.

(The reviewers’ comments are in Italics. Our responses are non-italicized)

Reviewer #2

  1. In lines 259-261, these data have suggested that TORC2 positively regulates cell growth and survival through Akt/FOXO signaling. In contrast, in line 284-285, these data indicate that the effects of TORC2 on cell growth are not mediated by its target Akt and FOXO signaling. There is an inconsistency between these two sentences. Please describe the author’s views on these points.

We have now amended the manuscript (Lines 278-280 and Lines 297-307) to clarify the inconsistency between the sentences.

  1. Lines 303-305, from the original expression, I feel that amino acids stimulation facilitates formation of Rag heterodimer, instead of the promotion of changes of GTP- GDP cycles of each Rag GTPases. Please rewrite.

We have rephrased the sentences (Lines 325-328) as follows:

In mammalian cells, the RagA/RagB or RagC/RagD stable heterodimers are recruited to the lysosomal membrane by the Regulator complex [130]. Upon amino acid stimulation, Rags are converted to their nucleotide-active state to mediate cellular translocation of the mTORC1 complex to the lysosomes [130].

  1. Lines 170-177, the authors should briefly summarize the molecular function of TBC1D7. Is TBC1D7 not required for TSC in Drosophila?

We have amended the manuscript to clarify the molecular function of TBC1D7 and its relationship with the TSC complex in Drosophila (Lines 175-181).

  1. Throughout the manuscript, the authors use mTORC1 and TORC1 (mTORC2 and TORC2). I understand that mTORC1 and mTORC2 may be used only for mammals. However, for example, in line 203, the authors used “mTORC1”, but it should be TORC1 because the original experimental data were obtained in Drosophila. Most readers might be confused. Please add the definition of how the terms are used, for example, mTORC1 for mammals, and TORC1 for other species such as Drosophila. Please confirm the use of these terms throughout the manuscript.

To facilitate the reading, we now use “mTORC1” and “mTORC2” as “mechanistic target of rapamycin complex 1” and “mechanistic target of rapamycin complex 2” respectively. Throughout the manuscript we use these terms for mammalian cells and all model organisms including Drosophila.

  1. Lines 55-69, Please modify.

We have now included the abbreviations in the Figure 1 Legend (Lines 55-67).

  1. Line 82, S6K1 is used, but S6K in Figure 1. Please modify the expression consistently.

We have now modified the Figure 1 Legend and the text to avoid inconsistency.

  1. Line104, full name of PIP2 and PIP3 should be --- bisphosphate and --- trisphosphate, respectively.

We have changed full name of PIP2 and PIP3 in the text (Lines 104-105).

  1. Line 108, PDPK1 should be PDK1.

We have corrected the typo (Line 110).
